# Dysbiosis of Oral Microbiota and Metabolite Profiles Associated with Type 2 Diabetes Mellitus

Yujiao Li,[a] Fei Qian,[b] Xiaogang Cheng,[a] Dan Wang,[a] Yirong Wang,[a] Yating Pan,[a] Liyuan Chen,[a] Wei Wang,[a] Yu Tian[a]

[a]State Key Laboratory of Military Stomatology & National Clinical Research Center for Oral Diseases & Shaanxi Key Laboratory of Stomatology & Department of Operative Dentistry and Endodontics, School of Stomatology, the Fourth Military Medical University, Xi'an, People's Republic of China

[b]State Key Laboratory of Military Stomatology & National Clinical Research Center for Oral Diseases & Shaanxi Key Laboratory of Stomatology & Department of Prosthodontics, School of Stomatology, the Fourth Military Medical University, Xi'an, People's Republic of China

Yujiao Li, Fei Qian, and Xiaogang Cheng contributed equally to this article. Author order was determined by their equal but gradated contributions for this paper.
Wei Wang and Yu Tian contributed equally to this article. Author order was determined by their equal but gradated contributions for this paper.

**ABSTRACT** Several previous studies have shown that oral microbial disorders may be closely related to the occurrence and development of type 2 diabetes mellitus (T2DM). However, whether the function of oral microorganisms and their metabolites have changed in patients with T2DM who have not suffered from any oral diseases has not been reported. We performed metagenomic analyses and nontargeted metabolic analysis of saliva and supragingival plaque samples from patients with T2DM who have not suffered any oral diseases and normal controls. We found that periodontal pathogens such as *Porphyromonas gingivalis* and *Prevotella melaninogenica* were significantly enriched, while the abundances of dental caries pathogens such as *Streptococcus mutans* and *Streptococcus sobrinus* were not significantly different in patients with T2DM compared to those in normal controls. Metabolomic analyses showed that the salivary levels of cadaverine and L-(+)-leucine of patients with T2DM were significantly higher than those of normal controls, while the supragingival plaque levels of *N*-acetyldopamine and 3,4-dimethylbenzoic acid in patients with T2DM were significantly higher than those in the normal controls. Additionally, we identified the types of oral microorganisms related to the changes in the levels of circulating metabolites, and the oral microorganisms were involved in the dysregulation of harmful metabolites such as cadaverine and *n*, *n*-dimethylarginine. Overall, our study first described the changes in the composition of oral microorganisms and their metabolites in patients with T2DM who have not suffered any oral diseases, which will provide a direct basis for finding oral biomarkers for early warning of oral diseases in T2DM.

**IMPORTANCE** The incidence of oral diseases in type 2 diabetic patients might increase, and the severity might also be more serious. At present, the relationship between oral microorganisms and type 2 diabetes mellitus (T2DM) has become a hot topic in systemic health research. However, whether the function of oral microorganisms and their metabolites have changed in patients with T2DM who have not suffered from any oral diseases has not been reported. We found that even if the oral condition of T2DM is healthy, their oral microbes and metabolites have changed, thus increasing the risk of periodontal disease. Our study first described the changes in the composition of oral microorganisms and their metabolites in T2DM who have not suffered any oral diseases and revealed the correlation between oral microorganisms and their metabolites, which will provide a direct basis for finding oral biomarkers for early warning of oral diseases in patients with T2DM.

**KEYWORDS** metagenomic, metabolomic, type 2 diabetes mellitus, oral diseases, oral microbiome

Address correspondence to Yu Tian, tianyu@fmmu.edu.cn, or Wei Wang, weiwang_0510@163.com.
The authors declare no conflict of interest.

Type 2 diabetes mellitus (T2DM) is a serious endocrine and metabolic disorder whose etiology is extremely complex. To date, its pathogenesis has not been elucidated clearly (1). The International Diabetes Federation (IDF) reported that the prevalence of T2DM has been annually increasing since 2000 (2). T2DM may lead to metabolic and hemodynamic disorders such as hyperglycemia and insulin resistance, which activate destructive processes (3). Recently, many studies have reported that there is a close relationship between T2DM and various oral diseases (4–7). The incidence of oral diseases in type 2 diabetic patients might increase, and the severity might also be more serious. T2DM reduces the quality of life of patients and increases the financial burden on individuals, families, and society.

Oral microorganisms, as one of the five major human bacterial communities, play an important role in the occurrence and development of dental caries, periodontal disease and other oral diseases via pathogen inhibition and immune regulation (8, 9). The oral microflora is a diverse and dynamic ecosystem in the human body, in which more than 700 species of bacteria have been detected (10). Oral microecological imbalance can cause oral diseases and is closely related to the occurrence and development of systemic diseases such as T2DM (11). Therefore, understanding the changes in oral microorganisms in patients with T2DM is helpful to explore the pathogenesis of oral disease in these patients. At present, studies of oral microorganisms in patients with T2DM are limited to 16S rDNA sequencing, which can sequence only a specific region of the microbial genome and cannot characterize the microbial community at the species level or functional metabolism (12). However, metagenomic sequencing can sequence the complete genome with higher resolution than 16S rDNA sequencing and improve the detection efficiency of microbial species, microbial diversity, and functional genes (13). At present, studies of oral microorganisms in type 2 diabetic patients who have not yet suffered from any oral diseases using metagenomic sequencing have not been reported. In addition, the metabolic activities of oral microorganisms are affected by changes in the oral microecological environment and can change the oral microecological environment, which may impact the specific selection of the environment for oral microorganisms, enhance the pathogenicity of bacteria, and eventually lead to the occurrence of oral diseases (14). Barnes et al. (15) used metabolomics to analyze the saliva of subjects with and without diabetes and found that compared with those of nondiabetic patients, the saliva samples of diabetic patients showed altered carbohydrate, lipid, and oxidative stress signatures. The metabolic profiling of oral microorganisms in patients with T2DM may reveal unique metabolic characteristics related to oral diseases in these patients. Some studies (16, 17) have shown that the analysis of saliva and plaque metabolites yields candidate metabolic biomarkers and pathways related to caries and periodontal disease, which may provide opportunities for the verification of early diagnostic models and new targets for the prevention and treatment of oral diseases and T2DM.

In short, metagenomic analysis is helpful to reveal the trend of changes in the microbial composition under different conditions, and metabolomic analysis can reveal the by-products produced in the oral microecological environment under different conditions (18). Although each method provides abundant information, the correlation between the changes in oral microorganisms and saliva and supragingival plaque metabolites in T2DM has not been reported, which will reveal valuable information on the relationship among the oral microbiome structure, metabolites, and T2DM. Therefore, this study included metagenomic and metabolomic analyses of saliva and supragingival plaque in patients with T2DM and first revealed the correlation among the composition, function, and metabolite profiles of the oral microbiota in type 2 diabetic patients. This study also revealed the relevance of changes in oral microbial species and several promising candidate metabolic biomarkers and pathways in type 2 diabetic patients who have not yet suffered from any oral diseases, which may be of great value for the early diagnosis, prevention, and treatment of oral diseases in patients with T2DM.

**TABLE 1** Demographic and clinical characteristics of all participants[a]

| Characteristics | T2DM patients | Healthy controls | P value |
|---|---|---|---|
| No. | 10 | 10 | _b |
| Male/female | 7/3 | 5/5 | 0.361 |
| Age (yrs) | 44.7 ± 11.67 | 42.20 ± 11.27 | 0.632 |
| ht (cm) | 1.71 ± 0.07 | 1.68 ± 0.08 | 0.422 |
| wt (kg) | 76.3 ± 11.72 | 61.80 ± 7.32 | 0.005* |
| BMI (kg/m²) | 25.97 ± 2.33 | 21.84 ± 1.82 | 0.000* |
| DMFT | 0 | 0 | - |
| GI | 0 | 0 | - |
| PLI | 0.80 ± 0.63 | 0.70 ± 0.48 | 0.696 |
| PD (mm) | 2.48 ± 0.12 | 2.47 ± 0.13 | 0.859 |
| BI | 0 | 0 | - |
| CAL (mm) | 0 | 0 | - |

[a]Values are expressed as the mean ± standard deviation. Significant differences are indicated by *, $P < 0.05$. BMI, body mass index.
[b]The dashes in the table indicate that the data is not applicable.

## RESULTS

**Clinical characteristics of subjects.** We consecutively recruited 10 T2DM patients (7 males and 3 females) and 10 healthy controls (5 males and 5 females). The clinical characteristics of all subjects are shown in Table 1. There were no significant differences in age distribution or sex between the two groups, while significant differences in body mass index (BMI) were noted. In this study, we selected healthy controls and type 2 diabetic patients who did not suffer from periodontal disease, dental caries or other oral diseases to reduce the influence of oral health status on oral microorganisms.

**Compositional and functional alteration of the saliva and supragingival plaque microbiota in T2DM.** We first compared saliva and supragingival plaque microbial alpha diversity between T2DM patients and healthy controls using the Shannon index (Fig. 1A, and B). We found that there was no significant difference in alpha diversity between the T2DM group and the healthy control group in saliva or supragingival plaque. These results showed that there was no significant difference in richness, evenness, or diversity of oral microorganisms between T2DM patients and normal controls.

According to beta diversity analysis based on Bray–Curtis distances between the T2DM group and healthy control group, the structures of the supragingival plaque microbiota in T2DM were significantly different from those in controls ($R = 0.131$, $P = 0.026$, Fig. 1D), but those of the salivary microbiota were quite similar ($R = 0.042$, $P = 0.217$, Fig. 1C). PCoA based on Bray–Curtis distances showed consistent results (Fig. S1 and Fig. 2).

To dissect the detailed taxonomic characteristics, LEfSe analysis of salivary and supragingival plaque microorganisms in the T2DM group and healthy control group was carried out. LEfSe analysis demonstrated that these biomarkers were significantly different between the two groups (Fig. 1E and F). Using a logarithmic LDA score cutoff of 2, we identified 31 biomarkers in saliva samples as key discriminants (Fig. 1E). *Treponema denticola*, *Aggregatibacter segnis*, *Rothia mucilaginosa*, *Fusobacterium nucleatum*, *Catonella morbi*, *Porphyromonas gingivalis*, *Treponema vincentii*, *Parvimonas micra,* and *Morococcus cerebrosus* were significantly overrepresented in the T2DM group, whereas *Prevotella aurantiaca* and *Lautropia mirabilis* were enriched in the healthy control group. At the same time, we identified 41 biomarkers in supragingival plaque as key discriminants (Fig. 1F). *Neisseria flavescens*, *Capnocytophaga granulosa*, *Prevotella melaninogenica*, *Capnocytophaga_sp__oral_taxon_329,* and *Capnocytophaga_sp__CM59* were significantly overrepresented in the T2DM group, whereas *Actinomyces massiliensis*, *Corynebacterium durum*, *Propionibacterium propionicum*, *Actinomyces johnsonii*, *Streptococcus sanguinis*, *Actinomyces naeslundii*, *Lautropia mirabilis*, *Rothia aeria,* and *Cardiobacterium hominis* were enriched in the healthy control group. These results showed a remarkable difference in oral microbiota composition between the T2DM group and the healthy control group. In addition, the ratio of *Firmicutes/Bacteroidetes* (F/B) in T2DM patients was higher than in healthy controls, but there was no statistical significance (0.374 ± 0.146 versus 0.318 ± 0.069, $P > 0.05$). In order to describe the taxonomic profiles of the microbial communities more intuitively,

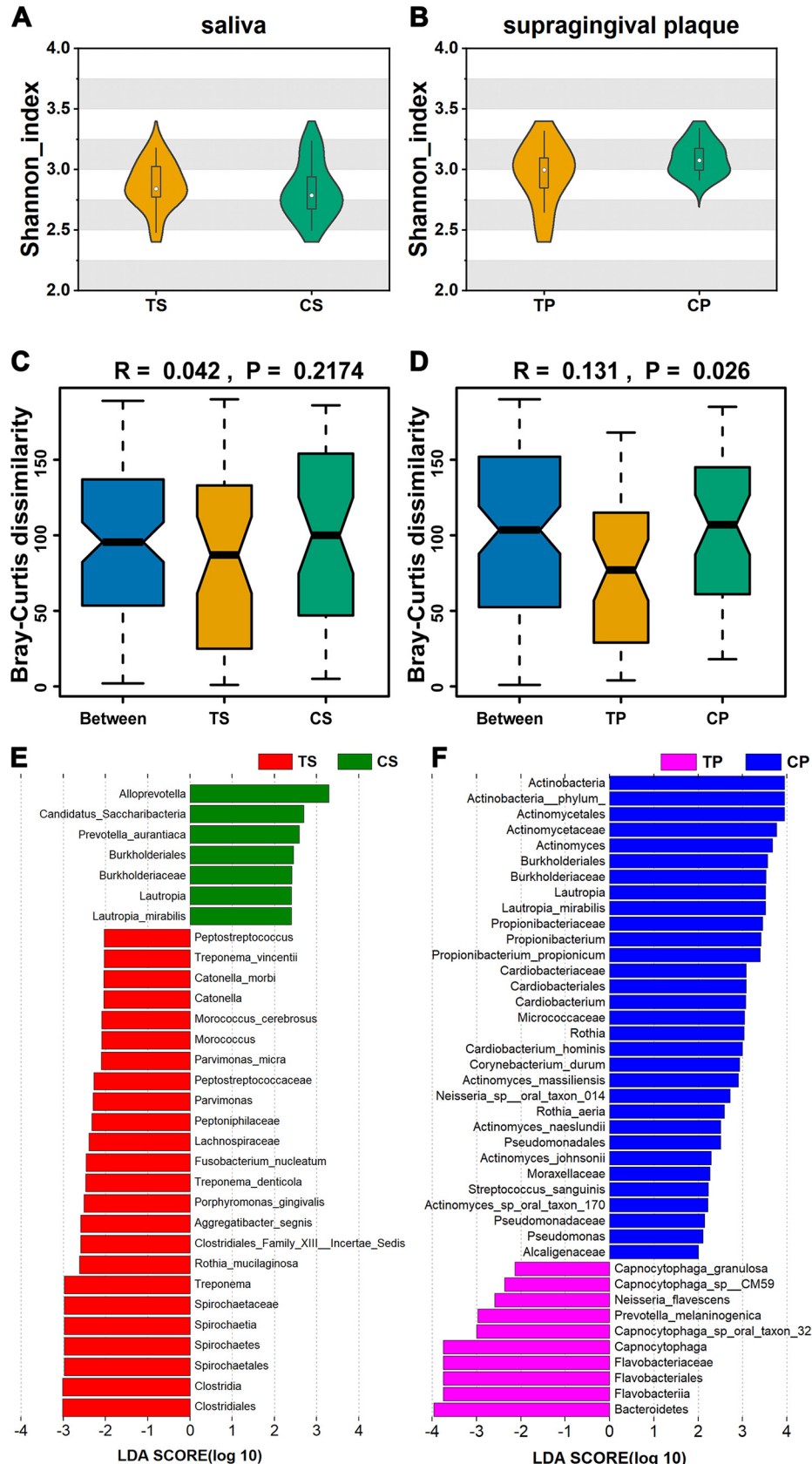

**FIG 1** Diversity analysis and taxonomic signatures of saliva and supragingival plaque microorganisms at the species level in the T2DM group and healthy control group. (A) Diversity analysis of saliva samples.

taxonomy barplots at genus level (Fig. 2A and C) and species level (Fig. 2B and D) were plotted for each sample. Top 30 taxonomies were plotted from a sorted abundance table when more than 30 taxonomies are annotated.

To explore the distinct functions of the oral microbiota between T2DM patients and healthy controls, we annotated the metagenomic function of KEGG modules. Based on the threshold values Reporterscore $> 1.65$, 32 KEGG pathways were significantly enriched in T2DM saliva, and 4 KEGG pathways were significantly enriched in control saliva (Fig. S3). Twenty-eight KEGG pathways were significantly enriched in T2DM supragingival plaque, and 36 KEGG pathways were significantly enriched in control supragingival plaque (Fig. S4).

**Saliva and supragingival plaque metabolomics in T2DM patients.** Considering the difference in metabolic functions between T2DM patients and controls based on metagenomic function annotations, we further used high-throughput liquid chromatography–mass spectrometry (LC/MS) to analyze the metabolic profiles of saliva and supragingival plaque samples. The saliva and supragingival plaque samples were subjected to LC/MS analysis in both positive ion mode (ES+) and negative ion mode (ES−). The PCA algorithm was used to observe the distribution and separation trend of metabolites in saliva and supragingival plaque samples of T2DM patients and normal controls (Fig. 3). As shown in Fig. 2, the distribution of metabolites in saliva (Fig. 3A and B) and supragingival plaque samples (Fig. 3C and D) between the T2DM group and the control group was significantly separated, indicating that there were metabolic differences between T2DM patients and normal controls. To further identify metabolites that discriminate between T2DM patients and healthy normal controls, we performed partial least-squares discriminant analysis (PLS-DA) on the metabolic data of saliva and supragingival plaque samples. Distinct separations were present between the T2DM and control groups in this model in both POS (Fig. 3E and F) and NEG (Fig. 3G and H) modes.

The combined thresholds variable importance in the projection (VIP) scores from the OPLS-DA model $>1$, fold change $\geq 1.2$ or $\leq 0.83$, and $P$ value $< 0.05$ were used to identify differentially abundant metabolites between T2DM and normal controls. Metabolites with VIP $> 1.5$ and $P < 0.05$ were selected as significantly differentially abundant metabolites. Finally, 1,455 differentially enriched metabolites in POS mode (Fig. 4A) and 399 differentially enriched metabolites in NEG mode (Fig. 4B) were identified in saliva samples of the T2DM and control groups. The differentially abundant metabolites were visualized by volcano plots. Among them, the levels of cadaverine, S-hexyl-glutathione, daidzein, L-(+)-leucine, and 37 other metabolites were significantly higher in the T2DM group than in the control group, while the levels of pentosidine, L-serine, and 25 other metabolites were significantly lower in the T2DM group than in the control group (Table S1). In addition, 136 differentially enriched metabolites in POS mode (Fig. 4C) and 34 differentially enriched metabolites in NEG mode (Fig. 4D) were identified in supragingival plaque samples from the T2DM and control groups. The differentially abundant metabolites were visualized by volcano plots. Among them, the levels of N, n-dimethylarginine, N-acetyldopamine, 3,4-dimethylbenzoic acid, and 58 other metabolites were significantly higher in the T2DM group than in the control group, while the levels of N-acetylvaline, sucrose, 2,2-bis(hydroxymethyl)propionic acid, and 14 other metabolites were significantly lower in the T2DM group than in the control group (Table S2). Based on the KEGG database, the metabolic pathways of differentially abundant metabolites in saliva were enriched and analyzed, among which metabolic pathways with $P$ values $<0.05$ were considered to exhibit a significant difference. Nine metabolic pathways (tryptophan metabolism, neomycin, kanamycin and gentamicin biosynthesis, and pyrimidine metabolism) in POS mode (Fig. 4E) and 10 meta-

**FIG 1** Legend (Continued)

(B) Diversity analysis of supragingival plaque samples. (C) ANOSIM of saliva samples. (D) ANOSIM of supragingival plaque samples. (E) LEfSe analysis of saliva samples. (F) LEfSe analysis of supragingival plaque samples. TS refers to the saliva samples of T2DM group. TP refers to the supragingival plaque samples of T2DM group. CS refers to the saliva samples of control group. CP refers to the supragingival plaque samples of control group.

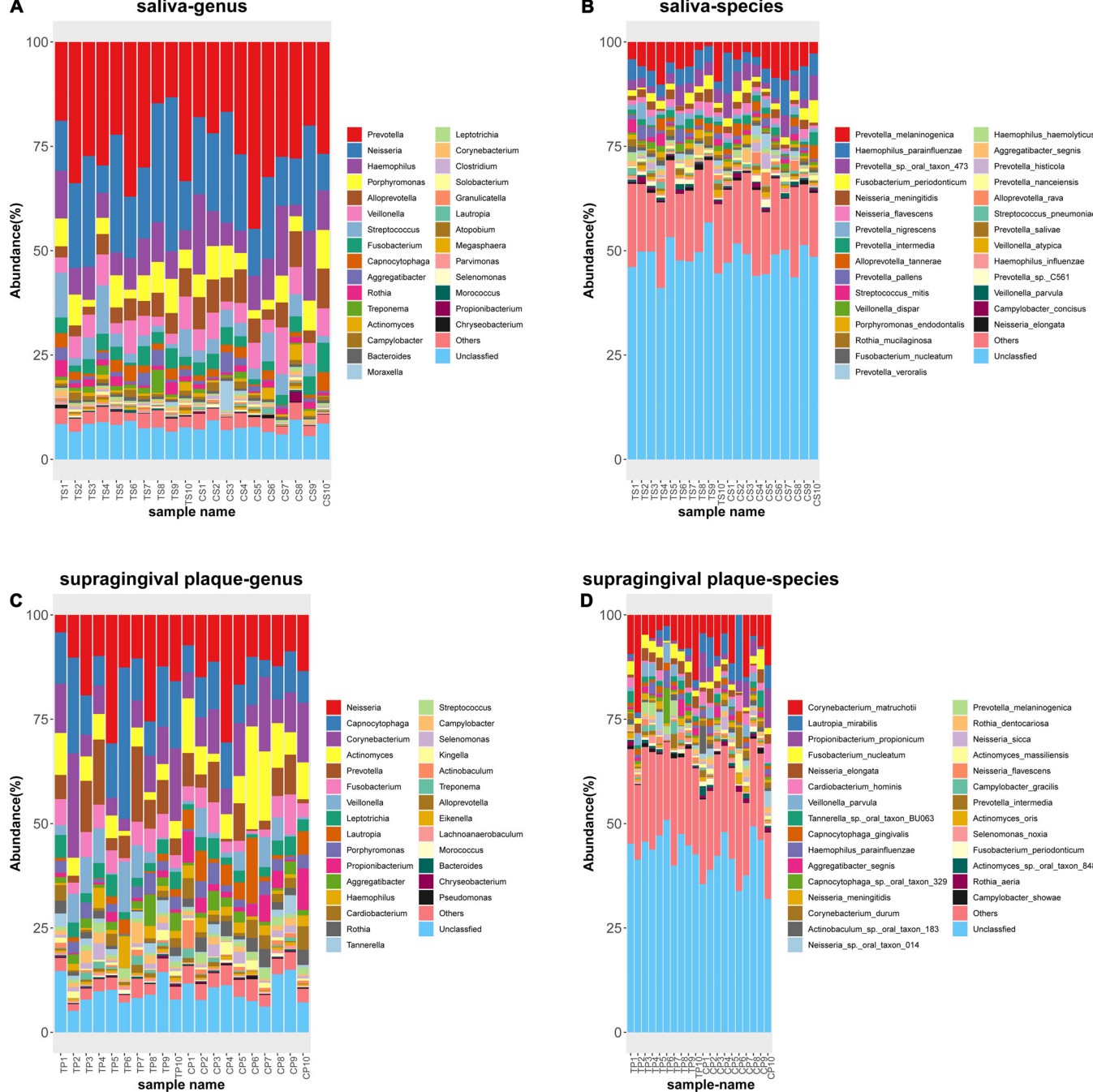

**FIG 2** Taxonomic profiles of the microbial communities at the genus level and species level in each sample. (A) T2DM salivary bacteria at the genus level; (B) Control salivary bacteria at the genus level; (C) T2DM supragingival plaque bacteria at the species level; (D) Control supragingival plaque bacteria at the species level. TS refers to the saliva samples of T2DM group. TP refers to the supragingival plaque samples of T2DM group. CS refers to the saliva samples of control group. CP refers to the supragingival plaque samples of control group.

bolic pathways (ABC transporters, purine metabolism and neomycin, kanamycin and gentamicin biosynthesis) in NEG mode (Fig. 4F) were significantly enriched (Table S3). Moreover, the metabolic pathways of differentially abundant metabolites of supragingival plaque samples were enriched and analyzed, among which metabolic pathways with $P$ value $<0.05$ were considered to exhibit a significant difference. Five metabolic pathways (tyrosine metabolism, vitamin digestion and absorption, and the AMPK signaling pathway) in POS mode (Fig. 4G) and 14 metabolic pathways (ABC transporters, pyrimidine metabolism, and phenylalanine metabolism) in NEG mode (Fig. 4H) were significantly enriched (Table S4).

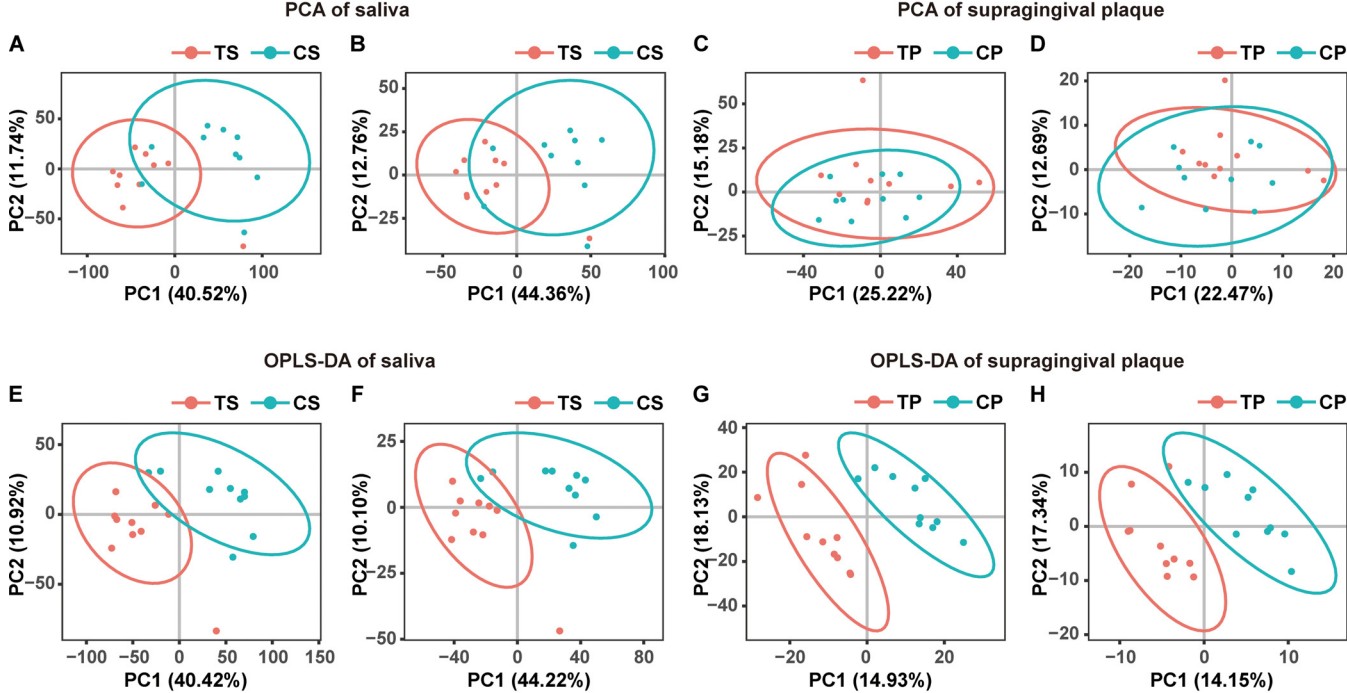

**FIG 3** Multivariate analysis of serum metabolites. (A) PCA of saliva (POS); (B) PCA of saliva (NEG); (C) PCA of supragingival plaque (POS); (D) PCA of supragingival plaque (NEG); (E) OPLS-DA of saliva (POS); (F) OPLS-DA of saliva (NEG); (G) OPLS-DA of supragingival plaque (POS); (H) OPLS-DA of supragingival plaque (NEG). TS refers to the saliva samples of T2DM group. TP refers to the supragingival plaque samples of T2DM group. CS refers to the saliva samples of control group. CP refers to the supragingival plaque samples of control group.

**Correlation analysis of the changes in metabolites and different microbial species.** To explore the functional correlation between the changes in oral microbial structure and the changes in metabolites in patients with T2DM, Spearman correlation analysis was carried out. In saliva samples, we analyzed possible correlations between 63 significantly altered metabolites (VIP > 1.5) and microbial species (30 differential bacterial species) based on Spearman's correlation (Table S5). Specifically, some periodontal pathogens such as *Parvimonas micra*, *Porphyromonas gingivalis*, and *Treponema denticola* were significantly associated with 15, 32, and 40 salivary metabolites, respectively ($P < 0.05$). In addition, L-serine was negatively correlated with *Parvimonas micra* and *Treponema denticola*; hexanoylcarnitine was negatively correlated with *Porphyromonas gingivalis*; and cadaverine was positively correlated with *Porphyromonas gingivalis* and *Treponema denticola*. The detailed Spearman's correlation coefficients and *P* values are shown in Table S5. In supragingival plaque samples, we analyzed possible correlations between 76 significantly altered metabolites (VIP > 1.5) and microbial species (30 differential bacterial species) based on Spearman's correlation (Table S5). Among them, the increased *Campylobacter concisus* abundance in T2DM was positively correlated with *n*, *n*-dimethylarginine levels.

## DISCUSSION

Oral microorganisms are closely related to the pathogenesis of oral diseases in patients with T2DM, and the metabolic activities of oral microorganisms also play an important role in the occurrence and development of oral diseases (18). However, there is still a lack of research on the combined analysis of the oral microbial metagenome and metabolome in patients with T2DM who have not yet suffered from any oral diseases. This is the first study to combine metagenomic sequencing with LC–MS-based metabolomics for saliva and supragingival plaque microflora in patients with T2DM. The results showed that there were significant differences in the microbial flora composition, function, and metabolites of saliva and supragingival plaque in patients with T2DM compared with those in normal controls.

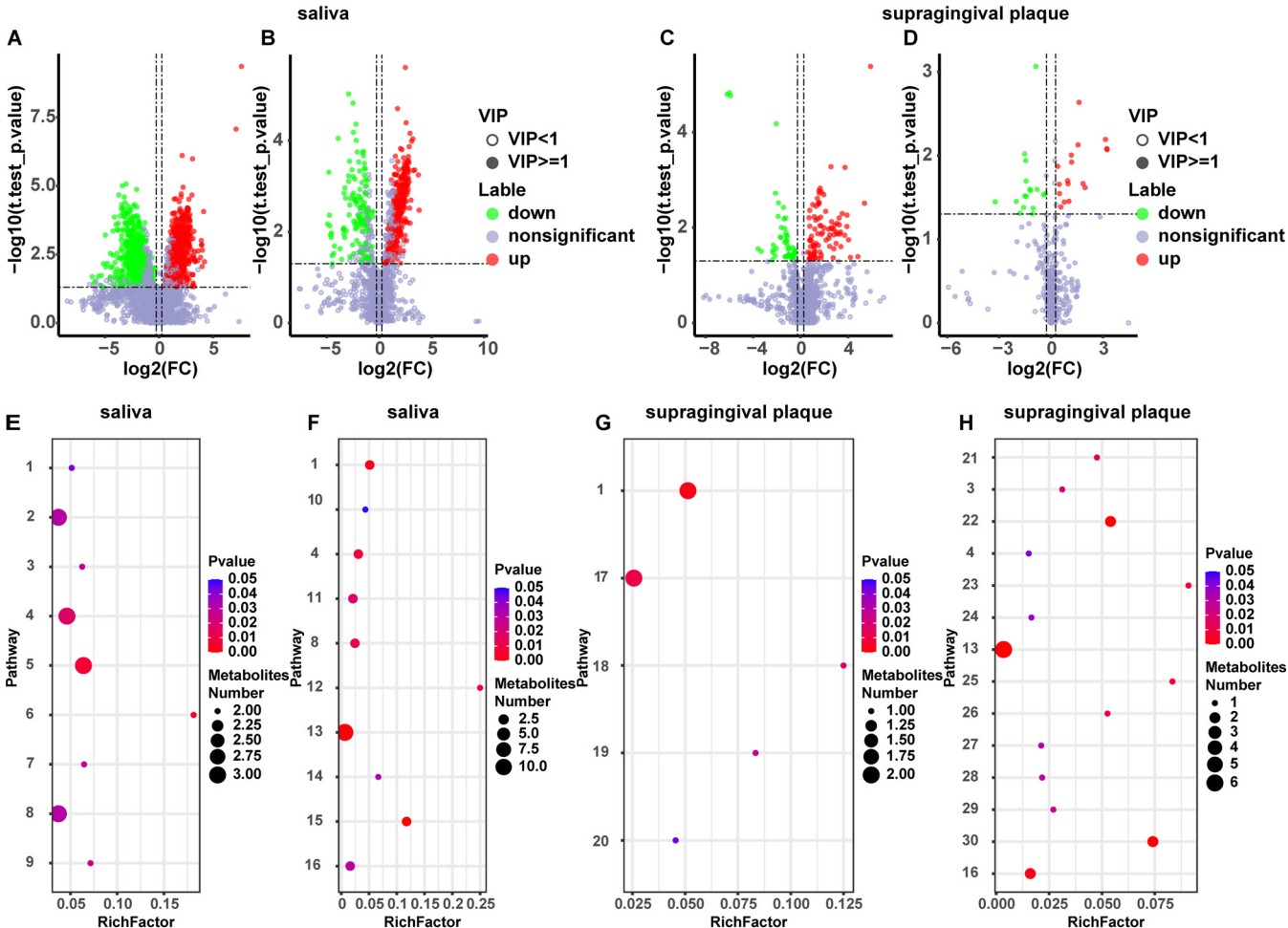

**FIG 4** Analysis of differentially abundant metabolites. (A) Metabolites of saliva (POS); (B) Metabolites of saliva (NEG); (C) Metabolites of supragingival plaque (POS); (D) Metabolites of supragingival plaque (NEG); (E) Metabolites of saliva (POS); (F) Metabolites of saliva (NEG); (G) Metabolites of supragingival plaque (POS); (H) Metabolites of supragingival plaque (NEG). (A–D) The x axis represents $\log_2$ (Fold Change); y axis represents the negative logarithmic conversion of the P-value of the difference test between the two groups (-$\log_{10}$ P-value). E~H: x axis enrichment factor (Rich Factor) is the number of differential metabolites annotated to the pathway divided by all identified metabolites annotated to the pathway. y axis represents pathways. 1 = Vitamin digestion and absorption; 2 = Tryptophan metabolism; 3 = Taste transduction; 4 = Pyrimidine metabolism; 5 = Protein digestion and absorption; 6 = Prostate cancer; 7 = Pathways in cancer; 8 = Neomycin, kanamycin and gentamicin biosynthesis; 9 = Biotin metabolism; 10 = Valine, leucine and isoleucine biosynthesis; 11 = Purine metabolism; 12 = mTOR signaling pathway; 13 = Metabolic pathways; 14 = HIF − 1 signaling pathway; 15 = Antifolate resistance; 16 = ABC transporters; 17 = Tyrosine metabolism; 18 = Longevity regulating pathway; 19 = Dopaminergic synapse; 20 = AMPK signaling pathway; 21 = Thyroid hormone synthesis; 22 = Starch and sucrose metabolism; 23 = Prolactin signaling pathway; 24 = Phenylalanine metabolism; 25 = Insulin secretion; 26 = Insulin resistance; 27 = Inositol phosphate metabolism; 28 = Galactose metabolism; 29 = Central carbon metabolism in cancer; 30 = Carbohydrate digestion and absorption.

Metagenomic sequencing has been applied to deeply analyze the relationship between microbial function and host physiological state. Our study showed that there was no significant difference in the alpha diversity of salivary and supragingival plaque microorganisms between the T2DM group and normal controls, and there was no significant difference in the beta diversity of salivary microorganisms, but there was a significant difference in the beta diversity of supragingival plaque microorganisms. The difference in saliva and supragingival plaque between the T2DM group and the control group may be due to the different physical and chemical properties of different oral microecological environments. Salivary microorganisms are mostly planktonic microorganisms, while supragingival plaque depends on biofilm growth. The synergism and antagonism within the biofilm may also mediate the colonization and distribution of different flora (19). The reports of Almeida-Santos (20) and Tam (21) also found that there was no significant difference in salivary microbial alpha diversity and beta diversity between patients with T2DM and normal controls at different species levels. However, Saeb et al. (22) reported that the species and phylogenetic diversity of the oral salivary microflora in T2DM patients were significantly lower than those in individuals

with normal blood glucose content. This was inconsistent with our findings possibly because the method they used was 16S rDNA sequencing, and the subjects they chose were not people with good oral health.

The ratio of *Firmicutes/Bacteroidetes* (F/B) plays an important role in maintaining the health of the intestinal tract (23). In order to verify the consistency between the intestinal and oral microflora found by recent studies (24, 25), our study examined the F/B ratio between T2DM and normal controls. We found that although the mean value of F/B ratio increased slightly in the T2DM group, there was no statistical difference. However, a study on the sequence of the V1-V2 region of the 16S rRNA gene showed that the F/B ratio and *Haemophilus* abundance among oral microbes in T2DM patients increased (26). We speculate that the reasons for the different conclusions may lie in the different samples selected, the differences in sequencing technology, or the fact that all the subjects in our study were in good oral health. Therefore, whether the ratio of F/B in oral microorganisms plays a role in the pathogenesis of oral diseases in patients with T2DM needs to be further verified. The results of LEfSe analysis showed that the abundances of periodontal pathogens such as *Porphyromonas gingivalis*, *Treponema denticola,* and *Fusobacterium nucleatum* in T2DM patients were significantly higher than those in normal controls. *Porphyromonas gingivalis* is a major oral pathogen that is closely related to periodontal diseases such as periodontitis and alveolar bone resorption (27), and a recent study showed that the glycosylation that occurs in T2DM might promote the formation of bacterial biofilms by increasing the content of heme in glycosylated hemoglobin, thus increasing the pathogenic potential of *Porphyromonas gingivalis* (28). Therefore, according to the change trend of oral microorganisms, we speculate that the potential risk of periodontal disease may be increased in T2DM, which will be further verified in our subsequent metabolomic results. In this study, common dental caries pathogens, such as *Streptococcus mutans*, *Lactobacillus,* and *Streptococcus sobrinus* in T2DM patients were not significantly different from those in normal subjects. Other studies have reported similar results. Collin (29) found that compared with that in the control group, the incidence of dental caries in the T2DM group was not significantly increased, and there was no significant difference in the abundance of acid-producing microorganisms such as *Streptococcus mutans* between the two groups. Therefore, our study preliminarily indicated that there was no significant difference in the risk of dental caries between T2DM patients and normal controls. We suspect that the reasons may be various. On the one hand, that restriction of ingestion of refined carbohydrates reduced caries in patients with T2DM. On the other hand, a high salivary glucose level (30) or hyposalivation (31) in T2DM might increase the caries risk. Therefore, it may be these contradictory associations that lead to no significant change in oral cariogenic bacteria in patients with T2DM. As for whether the actual incidence of dental caries in patients with type 2 diabetes will change, we believe that longitudinal trials with a larger sample size are needed to investigate.

Revealing the metabolomic characteristics of saliva and supragingival plaque microorganisms is helpful to deeply understand how microbial changes lead to the dysbiosis of the oral environment. Our study identified 19 metabolic pathways in the analysis of differentially abundant metabolites in saliva from patients with T2DM, of which the ABC transporter pathway was the most enriched, including 124 differentially abundant metabolites. We speculate that the ABC transporter pathway may be disrupted in the saliva of patients with T2DM. As one of the largest transmembrane protein superfamilies, ABC transporters are not only necessary for bacterial survival, virulence, and pathogenicity but also participate in the metabolic process of a variety of bacteria and the absorption of nutrients such as $Fe^{2+}$, amino acids, vitamins and oligopeptides, thus promoting the survival of bacteria in the host microenvironment (32). ABC transporter genes play a role in the pathogenesis of *Porphyromonas gingivalis*, as one of periodontal pathogens. Gao et al. have reported that the ABC transporter genes PG_RS04465 and PG_RS07320 played an important role in the infection of gingival epithelial cells

by *Porphyromonas gingivalis*. These results indicated that bacterial ABC transporters were essential for pathogenicity of *Porphyromonas gingivalis* (33). Therefore, we speculate that ABC transporters may be a clue in the pathogenesis of periodontal disease in patients with type 2 diabetes mellitus. However, there is still a lack of specific molecular mechanisms of ABC transporters in oral microorganisms of patients with T2DM, and further research can be carried out. In addition, 23 metabolites matched valine and leucine and isoleucine biosynthesis pathways, which were associated with a higher risk of T2DM (34). Our study also showed that cadaverine, S-hexyl-glutathione, daidzein, L-(+)-leucine, 3-(4-hydroxyphenyl) propionic acid, and *N*-acetyl-l-phenylalanine were significantly enriched in the saliva of patients with T2DM. Among them, the abundance of cadaverine in the saliva of patients with severe periodontitis was increased (15, 17), which was positively correlated with the accumulation of dental biofilm. Therefore, according to the above results, we speculated that even if the oral status of patients with T2DM was healthy, the microbes in their saliva may have led to the accumulation of biofilms and the production of cadaverine, which further increases the risk of periodontal disease. In the metabolic spectrum of supragingival plaque, 19 metabolic pathways with significant changes were found in supragingival plaque of T2DM patients. Interestingly, there were also 124 metabolites in supragingival plaque that matched the ABC transporter pathway, so we speculated that the ABC transporter pathway may play an important role in oral microecological disorders in T2DM. In addition, 78 metabolites matched the tyrosine metabolism pathway. The colonization and pathogenicity of *Porphyromonas gingivalis*, one of the periodontal pathogens, were regulated by signal transduction pathways based on protein tyrosine phosphorylation and dephosphorylation. In the presence of tyrosine, a tyrosine phosphatase designated Php1 can promote the development and pathogenicity of *Porphyromonas gingivalis*. And in the absence of tyrosine phosphatase Php1 activity *Porphyromonas gingivalis* was unable to cause periodontal disease in a mouse model of periodontitis (35). Notably, the related metabolites of ABC transporters and tyrosine metabolism need to be quantitatively analyzed by targeted metabolomics, and the role of these pathways in the pathogenesis of oral diseases in T2DM needs further study.

There is a significant correlation between the oral microflora and oral metabolites in T2DM, suggesting that disorders of the oral microflora are associated with metabolomic changes. Our study revealed that the significantly enriched periodontal pathogens in the salivary microflora of T2DM patients were correlated with some differentially abundant metabolites. Periodontal pathogens mainly include *Parvimonas micra*, *Porphyromonas gingivalis*, and *Treponema denticola*. The main differentially abundant metabolites were cadaverine, histidine, and L-serine. Previous studies have reported that cadaverine in the saliva of patients with periodontitis is significantly enriched, and the combination of cadaverine, 5-oxoproline, and histidine yielded satisfactory accuracy (area under the curve = 0.881) for the diagnosis of periodontitis (17). Cadaverine in saliva is positively correlated with the accumulation of dental biofilm. The increase in cadaverine content leads to damage to the dental epithelial barrier, which aggravates periodontitis (36). As an advanced glycation end product (AGE), pentosidine may promote oxidative stress and endothelial dysfunction (37) and inhibit the formation of bone nodules, resulting in bone fragility (38). *Campylobacter concisus* abundance increased significantly among the supragingival plaque microorganisms of T2DM patients and showed a positive correlation with the levels of the metabolite *n*, *n*-dimethylarginine. *N*, *n*-dimethylarginine is an endogenous inhibitor of endothelial nitric oxide (NO) synthase (39), and NO content is reduced in periodontal disease (40). *Campylobacter concisus* is an emerging bacterial pathogen that may play a role in the development of oral inflammatory conditions such as periodontal disease (41, 42), which may also be due to its inhibition of NO production during periodontal inflammation. These related data suggest that patients with type 2 diabetes have significant taxonomic disturbances in the saliva and supragingival plaque microflora, which may also lead to significant changes in related metabolic profiles. Therefore, based on the results of the correlation analysis of the above microorganisms and metabolites, we

hypothesized that the risk of periodontal disease in patients with T2DM increased. Moreover, cadaverine and *n,n*-dimethylarginine may become new potential markers for the prevention and treatment of periodontal disease in type 2 diabetic patients.

In our study, the BMI of T2DM group was higher than that of normal control group. Indeed, it has been reported that oral microbes may be associated with obesity (43, 44). However, previous studies have shown that the natural result of increased glucose and insulin availability was increased fat production and storage, thus, individuals who have T2DM are genetically predisposed to muscle insulin resistance, to become overweight or obese, the hallmark of type 2 diabetes mellitus (45). Hence, obesity may be one of the factors of T2DM to affect oral microbes, but we are still not sure whether obesity is independent of the effects of T2DM on oral microbes. In the future, we need more studies to explore the effects of obesity on oral microorganisms in patients with T2DM. Only in this way can we better clarify the pathogenesis of oral diseases in patients with T2DM. Our study provides preliminary evidence for the exploration of the pathogenesis of oral diseases in patients with T2DM. The relationship between oral metabolites and microorganisms and their role in the pathogenesis of oral diseases in patients with T2DM need to be further verified, and the causality and correlation between them still need further explanation and clarification.

**Conclusions.** Our study established a preliminary understanding of the relationship between the oral microflora at the species level and oral metabolites in patients with T2DM based on metagenomics and metabolomics. There were significant differences in the species composition, functional genes, and metabolism of oral microorganisms between T2DM patients and normal controls. It is also predicted that even type 2 diabetic patients with good oral health will have an increased risk of periodontal disease compared with that of normal controls. Future studies should longitudinally assess the microbial flora before and after the onset of T2DM and expand the sample size to better understand the pathogenesis of oral diseases, thus providing a theoretical basis for the biological prevention and treatment of oral diseases in patients with T2DM.

## MATERIALS AND METHODS

**Study subjects.** We collected saliva and supragingival plaque samples from a cross-sectional cohort of 20 subjects comprising 10 patients with T2DM and 10 healthy controls for shotgun metagenomic sequencing and untargeted metabolomics analyses, respectively. All T2DM subjects in this study were required to meet the following inclusion criteria: fasting plasma glucose (FPG)≥7.0 mmol/L, HbA1C ≥6.5%, and 2-h postprandial blood glucose (2-hPBG)≥11.1 mmol/L. The healthy subjects were recruited into this study after physical examination and evaluation. Participants were excluded if (i) they had used antibiotics or immunosuppressants in the past 6 months; (ii) they had used local antibiotics in the past 7 days; (iii) they had a history of HIV, HBV, or HCV infection; (iv) they were pregnant or lactating women; (v) they had cavities with caries, periodontal disease, oral abscess, oral mucosal disease, oral cancer, and so on; (vi) they had received oral scaling or periodontal treatment in the past 3 months; or (vii) they had more than eight missing teeth. An experienced dentist examined the oral cavity of the subjects in our study. The number of decayed, missing, or filled teeth (DMFT) was recorded with the exception of wisdom teeth. Gingival and periodontal tissues were examined in a logical order according to the Community Periodontal Index (46). First, the gingival tissues were examined with a visual inspection to assess (somewhat subjectively) the color and swelling of the tissues. Then, gingival index (GI), bleeding index (BI), clinical attachment loss (CAL), plaque index (PLI) and probing depth (PD) were measured by a KPC15 probe (Kangqiao, Shanghai, China).

**Sample collection.** All subjects were asked to avoid oral hygiene measures (such as brushing and flossing), eating, drinking, or gum chewing 12 h before sample collection. Saliva and supragingival plaque samples were collected from all the participants before breakfast. The mouths of all subjects were rinsed with pure water before samples were collected. Subjects expectorated at least 10 mL of unstimulated whole saliva into 50 mL sterile centrifuge tubes. Supragingival plaque samples were extracted from the necks of premolars and molars in the participants using sterile Gracey curettes and transferred into 1.5 mL of phosphate-buffered saline (PBS). The collected samples were placed in liquid nitrogen for flash freezing and then transferred within 2 h to a − 80℃ freezer in the laboratory for storage. Finally, 10 saliva samples (TS) and 10 supragingival plaque samples (TP) from patients with T2DM and 10 saliva samples (CS) and 10 supragingival plaque samples (CP) from controls qualified for metagenomic sequencing and untargeted metabolomic analysis.

**Metagenomic sequencing.** DNA extraction and purification were performed with a Qubit dsDNA BR assay kit (Invitrogen, USA) according to the manufacturer's instructions. The qualified libraries were sequenced on the MGISEQ-2000 platform (BGI-Shenzhen, China).

**Metagenomic data processing and analysis.** All the raw data were merged and trimmed by SOAPnuke v.1.5.2 (47). The pruned reads were mapped to the host genome using SOAP2 (48) software to identify and remove reads originating from the host. High-quality clean reads were *de novo* assembled by using IDBA-UD (49) software. In later calculations, assembled contigs with lengths less than 300 bp were removed. The genes of the over contigs were predicted by using MetaGeneMark (2.10) (50) software. Redundant genes were removed by using CD-HIT (51), and the recognition cutoff value was 95%. To generate taxonomic information, DIAMOND (52) was used to compare the protein sequence of the gene with the NR database, and the cutoff E value was 1e to 5. The MEGAN (53) LCA algorithm was used to assign the taxonomic annotation.

**Metabolomic analysis based on LC–MS.** After the saliva or supragingival plaque sample was thawed slowly at 4℃, 100 $\mu$L was placed in a 96-well plate, and 10 $\mu$L of internal standard was added to 300 $\mu$L of extract (methanol:I = 2:1, v:v, precooled at $-20$℃). After mixing by vortex for 1 min and standing at $-20$℃ for 2 h, saliva or supragingival plaque samples were centrifuged at 4,000 rpm and 4℃ for 20 min. After centrifugation, 300 $\mu$L of the supernatant was transferred for vacuum freeze drying. Then, 150 $\mu$L of complex solution (methanol:$H_2O$ = 2:1, v:v) was redissolved, vortexed for 1 min, and centrifuged at 4.000 rpm and 4℃ for 30 min. The supernatants were transferred to autosampler vials for LC–MS analysis.

**LC–MS analysis.** In this experiment, the metabolites were separated and detected by a Waters 2D UPLC (Waters, USA) tandem Q Exactive high-resolution mass spectrometer (Thermo Fisher Scientific, USA). The samples were analyzed on a Waters 2D UPLC (Waters, USA) coupled to a Q-Exactive mass spectrometer (Thermo Fisher Scientific, USA) with a heated electrospray ionization (HESI) source and controlled by the Xcalibur 2.3 software program (Thermo Fisher Scientific, Waltham, MA, USA). The chromatographic separation was implemented on a Waters ACQUITY UPLC BEH C18 column (1.7 $\mu$m, 2.1 mm $\times$ 100 mm, Waters, USA), and the temperature of the column was maintained at 45℃. The positive mode mobile phase was an aqueous solution containing 0.1% formic acid (liquid A) and 100% methanol (liquid B) containing 0.1% formic acid, and the negative mode mobile phase was an aqueous solution containing 10 mM formic acid ammonia (liquid A) and 95% methanol containing 10 mM formic acid ammonia (liquid B). The elution was carried out with the following gradient: 0 to 1 min, 2% liquid B; 1 to 9 min, 2% ~98% liquid B; 9 to 12 min, 98% liquid B; 12 to 12.1 min, 98% liquid B ~2% liquid B; and 12. 1 to 1 15 min, 2% liquid B. The flow rate was 0.35 mL/min, the column temperature was 45℃, and the injection volume was 5 $\mu$L. The sheath gas flow rate was 40 arbitrary units (arb), the auxiliary gas flow rate was 10 arb, the spray voltage in positive mode was 3.80 kV, the spray voltage in negative ion mode was 3.20 kV, the capillary temperature was 320℃, and the auxiliary gas heater temperature was 350℃. A Q Exactive mass spectrometer (Thermo Fisher Scientific, USA) was used to collect primary and secondary mass spectrometry data. The range of the mass-nucleus ratio of mass spectrometry scanning was 70 to 1050, the first-order resolution was 70,000, the AGC was 3e6, and the maximum injection time was 100 ms. According to the parent ion strength, the top 3 precursors were selected for subsequent MS/MS fragmentation with a resolution of 17,500, an AGC of 1e5, and a maximum ion injection time of 50 ms. The stepped normalized collision energy was set to 20, 40, and 60 eV. To provide more reliable experimental results during instrument testing, the samples were randomly sorted to reduce systematic errors. One QC sample was run for every 10 samples. All mass spectrometry raw data were imported into Compound Discoverer 3.1 (Thermo Fisher Scientific, USA) for data processing and feature extraction. Statistical analysis, metabolite classification annotations and functional annotations were performed using a self-developed metabolomics R package metaX (54) and metabolome bioinformatic analysis pipeline.

**Statistical analysis.** The clinical data conforming to a normal distribution are expressed as the mean $\pm$ standard deviation, and a *t* test was used for comparisons between groups. The clinical data with a nonnormal distribution are expressed as the median, and the Wilcoxon rank sum test was used for intergroup comparisons. All statistical tests in this study were bilateral tests, and $P < 0.05$ was considered significant. The data were statistically analyzed by SPSS (version 26.0). To obtain functional information, DIAMOND (52) was used to compare the protein sequences with the KEGG database (89.1), COG database (2014-11), eggNOG database (2015-10), Swiss-Prot database (2017-07), CAZy database (2017-09), and CARD database (4.0). The cutoff E value was 1e to 5. To generate taxonomic and functional abundance profiles, Bowtie 2 (55), which was applied with default settings, was used to align the reads with genes. Based on the abundance profiles, Wilcoxon's rank sum test (56) was used to determine the features (genera, phyla, and KOs) of significant differences in abundance among groups. The Benjamini-Hochberg (BH) (57) method was used to correct the $P$ value of multiple tests, and corrected $P$ values $< 0.05$ were considered significant. Differential enrichment of KEGG pathways was determined according to reporter scores (58). The absolute value of the reporter score was 1.65, which served as the significance detection threshold. Using relative abundance profiles at the gene, genus, and KO levels and the R package, the alpha diversity was quantified by the Shannon index. The Bray–Curtis distance (59) or Jensen–Shannon divergence distance (60) was used to calculate beta diversity. Principal-component analysis (PCA) was performed with the R software package "ade4." The R package VEGAN was used to perform principal coordinate analysis (PCoA). Spearman correlation coefficients between differentially enriched species and metabolites were calculated in R software (version 4.0.5).

**Ethics statement.** This study was conducted with subjects who had been tested for T2DM at the Xijing Hospital of the Air Force Medical University. The investigation conformed with the principles outlined in the Declaration of Helsinki. The samples and clinical information used in this study were obtained under conditions of informed consent and with approval of the institutional review board of the participating institute (Xijing Hospital of the Air Force Medical University, KY20212053-F-2). Written informed consent was obtained from all subjects.

**Data availability.** The data sets presented in our study can be found here: https://www.ncbi.nlm.nih
.gov/sra/PRJNA847441.

## SUPPLEMENTAL MATERIAL

Supplemental material is available online only.
**SUPPLEMENTAL FILE 1**, PDF file, 0.4 MB.
**SUPPLEMENTAL FILE 2**, XLSX file, 0.5 MB.
**SUPPLEMENTAL FILE 3**, XLSX file, 0.1 MB.
**SUPPLEMENTAL FILE 4**, XLSX file, 0.01 MB.
**SUPPLEMENTAL FILE 5**, XLSX file, 0.01 MB.
**SUPPLEMENTAL FILE 6**, XLSX file, 0.1 MB.

## ACKNOWLEDGMENTS

This work was supported by National Natural Science Foundation of China [grant no.
81970929].

The authors have no conflicts of interest to declare.

Y.L. designed and carried out part of the research, and conducted biological samples
collection and data analysis, as well as the writing of the original manuscript. F.Q.
carried out part of the research, analyzed and verified the data and wrote the original
draft of the manuscript. X.C. participated in study design, data collection and data
interpretation. D.W. did data analysis and interpretation. Y.W. conducted part of the
research work. Y.P. conducted biological samples collection. L.C. analyzed and verified
the data. W.W. provided supervision and guidance for research design and gave
detailed revision to the original manuscript. Y.T. provided supervision and guidance for
research design, data statistics and manuscript writing throughout the experiment, and
reviewed the manuscript. All authors made substantial contributions to the manuscript,
revised the manuscript critically, gave their final approval, and had full access to all the
data and accept responsibility to submit for publication.

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
