## [Reviewer comments · Microbiology Spectrum]

Microbiology Spectrum

Dysbiosis of oral microbiota and metabolite profiles associated with type 2 diabetes mellitus

Yujiao Li, Fei Qian, Xiao Cheng, Dan Wang, Yi Wang, Ya Pan, Li Chen, Wei Wang, and Yu Tian

Corresponding Author(s): Yu Tian, Department of Operative Dentistry and Endodontics, School of Stomatology, the Fourth Military Medical University

Review Timeline:

Submission Date:	September 19, 2022
Editorial Decision:	October 14, 2022
Revision Received:	November 7, 2022
Accepted:	December 12, 2022

Editor: Justin Kaspar

Reviewer(s): Disclosure of reviewer identity is with reference to reviewer comments included in decision letter(s). The following individuals involved in review of your submission have agreed to reveal their identity: Clifford J. Beall (Reviewer #1)

Transaction Report:

DOI: <https://doi.org/10.1128/spectrum.03796-22>

October 14, 2022

Prof. Yu Tian

Department of Operative Dentistry and Endodontics, School of Stomatology, the Fourth Military Medical University
Xi'an
China

Re: Spectrum03796-22 (Dysbiosis of oral microbiota and metabolite profiles associated with type 2 diabetes mellitus)

Dear Prof. Yu Tian:

The referenced manuscript was recently evaluated by two reviewers who provided critical feedback. Both reviewers commented on the lack of included detail on the study subjects -- both for the healthy controls and T2DM subjects. Both reviewers would like to see data included on the oral condition of the subjects (i.e. periodontal exams) and how obesity differences of the subjects was factored into the conclusions. Several other points were raised regarding caveats of conclusions drawn and missing figure legends and interpretation of supplementary data. If the authors feel they can address these issues, than a revised manuscript would be welcomed. A revised manuscript returned that does not include additional subject information would not be favorably viewed.

Link Not Available

Sincerely,

Justin Kaspar

Journals Department
Reviewer comments:

Reviewer #1 (Comments for the Author):

This paper describes shotgun metagenomic sequencing and metabolomics of subjects with type 2 diabetes and healthy controls. The authors studied 10 subjects of each group, a small number which means that the study can be regarded as a pilot. The small sample size is no doubt a trade off for the expense and technical difficulties of making metagenomic libraries.

The two sample types used were saliva and supragingival plaque, although subgingival plaque seems like it might be a more logical choice because of the known relationship of diabetes and chronic periodontitis.

The paper has a number of issues that need addressing, especially how it draws conclusions and represents previous work. I would urge the authors to carefully go through the manuscript and make sure that the references they are citing really support their claims.

Major points

The diabetes group was significantly more obese than the controls. Therefore the authors should include the strong caveat that all effects that they observe could be due to obesity equally as diabetes.

The authors claim that the T2DM subjects had no oral disease (line 23-24, 35, 46, 49, 73-75, 97, 214, 314) but I don't see the results of periodontal exams. The T2DM group has elevated *Treponema denticola* and *Porphyromonas gingivalis* in their saliva, presumably those are also elevated in subgingival plaque, so it certainly looks like they are likely to have some degree of periodontitis or gingivitis. If the perio exam wasn't done, those claims in the paper should be modified.

I don't think the correlation analysis in Figure 4 is worth including with 10 subjects per group and hundreds of bacteria and metabolites. That should be removed.

Lines 334-335: *Bacteroides* is a genus and they are not common in the oral cavity, Firmicutes is a phylum. Assuming you meant Bacteroidetes, there isn't any data presented in the paper on this ratio, not sure if it was accidentally left out.

Fusobacterium nucleatum is not a significant periodontal pathogen (see Griffen et al. and Abusleme et al.) though it is an opportunistic pathogen in other contexts.

I'm not convinced about ABC transporters being important based on some list of metabolites. Is that list just anything that can be transported by ABC transporters? Because that could be almost any small molecule, couldn't it? Sometimes these categories get a bit too broad to be useful.

Lines 382-385 "In the presence of tyrosine...[47]" Reference 47 doesn't mention tyrosine at all.

Lines 402-404: You say "N,n-dimethylarginine is an endogenous inhibitor of endothelial nitric oxide (NO) synthase, and NO content is reduced in periodontal disease [51]" but reference 51 doesn't mention periodontal disease.

Lines 404-406: You say: "*Campylobacter concisus* is a pathogen of periodontal disease and increases in abundance in periodontal disease, which may also be due to its inhibition of NO production during periodontal inflammation [52]" I assume you're talking about *C. concisus* but reference 52 is mainly about *C. jejuni*, *C. concisus* is only mentioned in passing and that reference doesn't deal with periodontitis or the oral cavity at all. Also *C. concisus* is not associated with periodontitis per Griffen et al. and Abusleme et al.

Figure 1 legend what are the y axis on C and D?

Figure 3 legend what are the x and y axis of the graphs?

I didn't get any legends for the supplemental figures

Please provide a link to the analysis code used

Minor points

Line 138: I believe the program is MetaGeneMark, not MetaGeneMarker

Line 189: the program is Bowtie2, not Botwie2

Lines 230, 231: where you say "species level branches" and "bacteria", what the figure is depicting is actually taxonomic ranks that can be different levels species or above

Line 241: not "fecal" bacteria

Lines 245 and 246 and figures - those abbreviations TS, CS, etc are never defined and I think it would be easier to understand with T2DM saliva or control saliva written out

Griffen, Ann, L, Beall, Clifford, J, Campbell, James, H, Firestone, Noah, D, Kumar, Purnima, S, Yang, Zamin, K, Podar M, Leys, Eugene, J. Distinct and complex bacterial profiles in human periodontitis and health revealed by 16S pyrosequencing. ISME J [Internet]. 2012 Jun;6(6):1176-85. Available from: <http://www.ncbi.nlm.nih.gov/pubmed/22170420>

Abusleme L, Dupuy AK, Dutzan N, Silva N, Burlinson JA, Strausbaugh LD, Gamonal J, Diaz PI. The subgingival microbiome in health and periodontitis and its relationship with community biomass and inflammation. ISME J [Internet]. 2013;7(5):1016-25. Available from: <http://www.ncbi.nlm.nih.gov/pubmed/23303375>

Reviewer #2 (Comments for the Author):

General comment

The authors examined saliva and supragingival plaque samples using metagenomic and metabolic analyses and found the differences in oral bacterial and metabolite compositions of patients with T2DM from those of healthy control. Particularly, it is interesting that the focus is on subjects who have not suffered any oral diseases. The present study could provide new

perspectives for understanding the involvement of diabetes in increasing risk of periodontal disease.

My major concern is the lack of details on oral conditions of subjects. The authors emphasize that the compositions of oral microbiota and metabolites change in T2DM patients even if the oral condition is healthy, and thus T2DM increases the risk of periodontal disease. However, the criteria of "healthy oral condition" in the present study is quite ambiguous. The authors only describe that subjects with caries or periodontal diseases are excluded, on line 114-116. The oral conditions are known to greatly influence on bacterial composition of oral microbiota. For instance, deepening of the periodontal pocket increases anaerobic environment in oral cavity and enriches anaerobic bacteria, even if the periodontal condition is within a healthy range. Therefore, it is possible that the present results just reflect differences in periodontal conditions rather than T2DM. In fact, the present results (increases of *P. gingivalis* or cadaverine in T2DM) are closely similar to alteration in oral cavity of subjects with deep periodontal pockets. Thus, it seems difficult to conclude T2DM increases periodontal disease risk via alteration of oral microbiota from the present results.

Specific comment

Result: To make it easier for the reader to understand the overall bacterial composition at genus or species level, the author should additionally draw figures showing bacterial composition in each subject (e.g. barplot).

Lines 276-307: It is hard to interpret these results from figure 3-4 and supplementary tables. For instance, which is the result of cadaverine? The authors should redraw figures to clarify it.

Lines 298-300: This sentence is unclear. Please rephrase.

Lines 354-358: These sentences are complicated. If a carbohydrate limited-diet is less likely to cause dental caries than a normal diet as described, doesn't the risk of dental caries decrease in T2DM patients?

Table S1, S2, and S5: Please show compound names.

Staff Comments:

Preparing Revision Guidelines

Please return the manuscript within 60 days; if you cannot complete the modification within this time period, please contact me. If you do not wish to modify the manuscript and prefer to submit it to another journal, please notify me of your decision immediately so that the manuscript may be formally withdrawn from consideration by Microbiology Spectrum.

Point-by-point response to the reviewers' comments

Editor's comments:

The referenced manuscript was recently evaluated by two reviewers who provided critical feedback. Both reviewers commented on the lack of included detail on the study subjects -- both for the healthy controls and T2DM subjects. Both reviewers would like to see data included on the oral condition of the subjects (i.e. periodontal exams) and how obesity differences of the subjects was factored into the conclusions. Several other points were raised regarding caveats of conclusions drawn and missing figure legends and interpretation of supplementary data. If the authors feel they can address these issues, than a revised manuscript would be welcomed. A revised manuscript returned that does not include additional subject information would not be favorably viewed.

Reply: We appreciate editor and reviewers very much for their positive and constructive comments on our manuscript. We have considered the comments very carefully and revised the paper accordingly. In this revised version, we have addressed the concerns of the reviewers. A point-by-point response to the reviewers' comments is enclosed. Thanks to the reviewers, we believe that this revised paper has been improved considerably. We hope that these revisions successfully address their concerns and requirements. Looking forward to hearing from you soon.

Note: In the response letter, all the responses are shown in blue, all the changes are highlighted in yellow, and all the reviewer's comments/suggestions are shown in black. Also, in the revised paper, all the changes and additions are highlighted in yellow.

Reviewer comments:

Reviewer #1 (Comments for the Author):

1. This paper describes shotgun metagenomic sequencing and metabolomics of subjects with type 2 diabetes and healthy controls. The authors studied 10 subjects of each group, a small number which means that the study can be regarded as a pilot. The small sample size is no doubt a trade off for the expense and technical difficulties of making metagenomic libraries.

Reply: We appreciate the reviewer's valuable comments. As the reviewer's comments said, we have to admit that the sample size of this study is limited. The reasons are as follows: first of all, as the reviewer's comments said, "*The small sample size is no doubt a trade off for the expense and technical difficulties of making metagenomic libraries*". Secondly, the inclusion criteria of the subjects in our study are strict, and the requirements for oral health status are relatively high, so it is not easy to collect patients with type 2 diabetes who meet the inclusion criteria. We sincerely hope that, as the reviewer said, "*...the study can be regarded as a pilot*" to provide some theoretical basis for exploring the relationship between type 2 diabetes and oral diseases in the future.

2. The two sample types used were saliva and supragingival plaque, although subgingival plaque seems like it might be a more logical choice because of the known relationship of diabetes and chronic periodontitis.

Reply: Thanks a lot for the reviewer's valuable comments. Generally speaking, saliva, supragingival plaque and subgingival plaque are selected in the study of oral microorganisms. However, considering that the subjects in this study are periodontal healthy, the subgingival space for bacterial growth may be limited when the periodontal tissue is healthy, so we finally decided to select saliva and supragingival plaque samples.

3. The paper has a number of issues that need addressing, especially how it draws conclusions and represents previous work. I would urge the authors to carefully go through the manuscript and make sure that the references they are citing really support their claims.

Reply: Thanks a lot for the reviewer's valuable comments. By reviewing the literature, some inappropriate conclusions and references have been corrected accordingly. Thanks to the reviewer, we believe that this revised paper has been improved considerably. We hope that these revisions successfully address your concerns and requirements.

Major points

4. The diabetes group was significantly more obese than the controls. Therefore the authors should include the strong caveat that all effects that they observe could be due to obesity equally as diabetes.

Reply: We appreciate the reviewer's valuable comments, it makes sense to take obesity into account in our study. In our study, the BMI of T2DM group was higher than that of normal control group. Indeed, it has been reported that oral microbes may be associated with obesity[58,59]. However, previous studies have shown that the natural result of increased glucose and insulin availability was increased fat production and storage, thus, individuals who have T2DM are genetically predisposed to muscle insulin resistance, to become overweight or obese, the hallmark of type 2 diabetes mellitus[60]. Hence, obesity may be one of the factors of T2DM to affect oral microbes, but we are still not sure whether obesity is independent of the effects of T2DM on oral microbes. In the future, we need more studies to explore the effects of obesity on oral microorganisms in patients with T2DM. Only in this way can we better clarify the pathogenesis of oral diseases in patients with T2DM. In the *Discussion* section of the manuscript, we have added a discussion of obesity. (see highlight at the end of the *Discussion* section in line 440-448).

Line 440-448 : “In our study, the BMI of T2DM group was higher than that of normal control group. Indeed, it has been reported that oral microbes may be associated with obesity[58,59]. However, previous studies have shown that the natural result of increased glucose and insulin availability was increased fat production and storage, thus, individuals who have T2DM are genetically predisposed to muscle insulin resistance, to become overweight or obese, the hallmark of type 2 diabetes mellitus[60]. Hence, obesity may be one of the factors of T2DM to affect oral microbes, but we are still not sure whether obesity is independent of the effects of T2DM on oral microbes. In the future, we need more studies to explore the effects of obesity on oral microorganisms in patients with T2DM. Only in this way can we better clarify the pathogenesis of oral diseases in patients with T2DM.”

[58]. Yang Y, Cai Q, Zheng W, Steinwandel M, Blot WJ, Shu XO, et al. Oral microbiome and obesity in a large study of low-income and African-American populations. *J Oral Microbiol.* 2019; 11: 1650597.

[59]. Wu Y, Chi X, Zhang Q, Chen F, Deng X. Characterization of the salivary microbiome in people with obesity. *PeerJ.* 2018; 6: e4458.

[60]. Malone JJ, Hansen BC. Does obesity cause type 2 diabetes mellitus (T2DM)? Or is it the opposite? *Pediatr Diabetes.* 2019; 20: 5-9

5. The authors claim that the T2DM subjects had no oral disease (line 23-24, 35, 46, 49, 73-75, 97, 214, 314) but I don't see the results of periodontal exams. The T2DM group has elevated *Treponema denticola* and *Porphyromonas gingivalis* in their saliva, presumably those are also elevated in subgingival plaque, so it certainly looks like they are likely to have some degree of periodontitis or gingivitis. If the perio exam wasn't done, those claims in the paper should be modified.

Reply: Thanks a lot for the reviewer’s valuable comments and suggestions. The comments and suggestions are very helpful for revising and improving our manuscript, as well as the important guiding significance to our study. The subjects we chose were actually oral health. An experienced dentist examined the oral cavity of the subjects in our study. The number of decayed, missing, or filled teeth (DMFT) were recorded with the exception of wisdom teeth. Gingival and periodontal tissues were examined in a logical order according to the Community Periodontal Index[19]. First, the gingival tissues were examined with a visual inspection to assess (somewhat subjectively) the color and swelling of the tissues. Then, gingival index (GI), bleeding index (BI), clinical attachment loss (CAL), plaque index (PLI) and probing depth (PD) were measured by a KPC15 probe (Kangqiao, Shanghai, China). The relevant contents in the *Materials and methods* section have been added accordingly in the revised manuscript. (see highlight at the end of the *Materials and methods* section in line 117-123). The relevant results have been added accordingly in Table 1.

Line 117-123: “An experienced dentist examined the oral cavity of the subjects in our study. The number of decayed, missing, or filled teeth (DMFT) was recorded with the exception of wisdom teeth. Gingival and periodontal tissues were examined in a logical order according to the Community Periodontal Index[19]. First, the gingival tissues were examined with a visual inspection to assess (somewhat subjectively) the color and swelling of the tissues. Then, gingival index (GI), bleeding index (BI), clinical attachment loss (CAL), plaque index (PLI) and probing depth (PD) were measured by a KPC15 probe (Kangqiao, Shanghai, China).”

[19].Eke PI, Page RC, Wei L, Thornton-Evans G, Genco RJ. Update of the case definitions for population-based surveillance of periodontitis. *J Periodontol.* 2012; 83: 1449-54.

Table 1:

Table 1 Demographic and clinical characteristics of all participants

Characteristics	T2DM patients	Healthy controls	p value
Number	10	10	-
Male/female	7/3	5/5	0.361
Age (years)	44.7 ± 11.67	42.20 ± 11.27	0.632
Height (cm)	1.71 ± 0.07	1.68 ± 0.08	0.422
Weight (kg)	76.3 ± 11.72	61.80 ± 7.32	*0.005
BMI (kg/m ²)	25.97 ± 2.33	21.84 ± 1.82	*0.000
DMFT	0	0	-
GI	0	0	-
PLI	0.80±0.63	0.70±0.48	0.696

PD(mm)	2.48 ±0.12	2.47±0.13	0.859
BI	0	0	-
CAL(mm)	0	0	-

Values are expressed as the mean ± standard deviation. Significant differences are indicated by * $p < 0.05$.

BMI, body mass index.

6.I don't think the correlation analysis in Figure 4 is worth including with 10 subjects per group and hundreds of bacteria and metabolites. That should be removed.

Reply: We appreciate the reviewer's valuable comments. Figure 4 has been removed accordingly in the revised manuscript.

7.Lines 334-335: *Bacteroides* is a genus and they are not common in the oral cavity, *Firmicutes* is a phylum. Assuming you meant *Bacteroidetes*, there isn't any data presented in the paper on this ratio, not sure if it was accidentally left out.

Reply: We appreciate the reviewer's valuable comments. The relevant contents have been rephrased as follows:

(1) The "*Bacteroides*" has been corrected to "*Bacteroidetes*". (see highlight in 346).

(2) The relevant F/B contents in the *Results* section have been added in the revised manuscript. (see highlight in Line 250-252).

Line 250-252 : "In addition, the ratio of *Firmicutes/Bacteroidetes* (F/B) in T2DM patients was higher than in healthy controls, but there was no statistical significance (0.374 ± 0.146 vs. 0.318 ± 0.069 , $P > 0.05$)."

(3) The relevant F/B contents in the *Discussion* section have been added in the revised manuscript. (see highlight in Line 346-355).

Line 346-355 : "The ratio of *Firmicutes/Bacteroidetes* (F/B) plays an important role in maintaining the health of the intestinal tract[38]. In order to verify the consistency between the intestinal and oral microflora found by recent studies[39,40], our study examined the F/B ratio between T2DM and normal controls. We found that although the mean value of F/B ratio increased slightly in the T2DM group, there was no statistical difference. However, a study on the sequence of the V1-V2 region of the 16S rRNA gene showed that the F/B ratio and *Haemophilus* abundance among oral microbes in T2DM patients increased[41]. We speculate that the reasons for the different conclusions may lie in the different samples selected, the differences in sequencing technology, or the fact that all the subjects in our study are oral health. Therefore, whether the ratio of F/B in oral microorganisms plays a role in the pathogenesis of oral diseases in patients with T2DM needs to be further verified."

8.*Fusobacterium nucleatum* is not a significant periodontal pathogen (see Griffen et al. and Abusleme et al.)

though it is an opportunistic pathogen in other contexts.

Reply: Thanks a lot for the reviewer's valuable comments. Thank you for providing us with two references. We have read them carefully.

In Griffen's article, *Fusobacterium nucleatum* was only mentioned casually in "Figure 7 Comparison of community profiles from the V1-2 and V4 variable regions", there is no additional discussion on *Fusobacterium nucleatum* in the main text. In Abusleme's article, "These OTUs may have important roles in subgingival communities providing structural support, as suggested for *Fusobacterium nucleatum*, serving as metabolic cornerstones or perhaps directing health to disease transitions." It seems that the relationship between *Fusobacterium nucleatum* and periodontitis was not discussed in detail in this article. However, in other studies, *Fusobacterium nucleatum* has been proved to be a periodontal pathogen. By carefully consulting the literature about *Fusobacterium nucleatum*, the reason why *Fusobacterium nucleatum* was used as a periodontal pathogen in our manuscript was explained as follows:

Fusobacterium nucleatum (*F. nucleatum*) is one of the most abundant species in the oral cavity. It is implicated in various forms of periodontal diseases including the mild reversible form of gingivitis and the advanced irreversible forms of periodontitis including chronic periodontitis, localized aggressive periodontitis and generalized aggressive periodontitis^[1, 2]. *F. nucleatum*, as a bridging bacterium, transfers critical periodontal pathogens to periodontal infectious sites and recruits and activates local immune cells, which results in tooth-supporting tissue destruction. And *F. nucleatum* has been identified as a high-frequency pathogen in periodontal disease^[3]. Animal studies supported a causative role of *F. nucleatum* in periodontal infections. Monoinfection of mice with *F. nucleatum* induced periodontal bone loss or abscess^[4]. There is a synergistic effect in the induction of experimental periodontitis by *P. gingivalis* and *F. nucleatum*. Polymicrobial infection with *P. gingivalis*/*F. nucleatum* aggravates alveolar bone loss and induces a stronger inflammatory response^[5]. In addition, a recent study also suggested that *F. nucleatum* inhibited gingival fibroblasts proliferation and promoted cell apoptosis, reactive oxygen species (ROS) generation, and inflammatory cytokine production partly by activating the AKT/MAPK and NF- κ B signaling pathways^[6]. In conclusion, based on the above theory, *Fusobacterium nucleatum*, as a periodontal pathogen, was detected in oral microorganisms of patients with type 2 diabetes mellitus in our study.

9. I'm not convinced about ABC transporters being important based on some list of metabolites. Is that list just anything that can be transported by ABC transporters? Because that could be almost any small molecule, couldn't it? Sometimes these categories get a bit too broad to be useful.

Reply: We appreciate the reviewer's valuable comments. We agree with the reviewer's comments. In most organisms, ABC transporters constitute one of the largest families of membrane proteins. In humans, ABC

transporters are implicated in a wide array of developmental processes, their functions are diverse and underpin numerous key physiological processes, as well as being causative factors in a number of clinically relevant pathologies^[7]. However, previous studies have found that atypical ABC proteins named sulfonylurea receptor (SUR1/ABCC8; SUR2/ABCC9) could generate ATP-sensitive potassium channel by forming a hetero-octameric complex with potassium channel Kir6.2 subunits, whose activity is critical for insulin secretion^[8]. In addition, ABC transporter gene plays a role in the pathogenesis of *Porphyromonas gingivalis*, as one of periodontal pathogens. Gao et al. have reported that the ABC transporter genes, PG_RS04465 and PG_RS07320 played an important role in the infection of gingival epithelial cells by *Porphyromonas gingivalis*. These results indicated that bacterial ABC transporters were essential for pathogenicity of *Porphyromonas gingivalis*^[9]. Therefore, we speculate that ABC transporters may be a clue in the pathogenesis of periodontal disease in patients with type 2 diabetes mellitus. However, there is still a lack of specific molecular mechanisms of ABC transporters in oral microorganisms of patients with T2DM in our study, and further research can be carried out. According to the reviewer's suggestion, the relevant contents have been rephrased accordingly (see highlight at the line 387-394)

Line 387-394 "ABC transporter genes play a role in the pathogenesis of *Porphyromonas gingivalis*, as one of periodontal pathogens. Gao et al. have reported that the ABC transporter genes, PG_RS04465 and PG_RS07320 played an important role in the infection of gingival epithelial cells by *Porphyromonas gingivalis*. These results indicated that bacterial ABC transporters were essential for pathogenicity of *Porphyromonas gingivalis*[48]. Therefore, we speculate that ABC transporters may be a clue in the pathogenesis of periodontal disease in patients with type 2 diabetes mellitus. However, there is still a lack of specific molecular mechanisms of ABC transporters in oral microorganisms of patients with T2DM, and further research can be carried out."

[48]. Gao L, Ma Y, Li X, Zhang L, Zhang C, Chen Q, et al. Research on the roles of genes coding ATP-binding cassette transporters in *Porphyromonas gingivalis* pathogenicity. *J Cell Biochem.* 2020; 121: 93-102.

10.Lines 382-385 "In the presence of tyrosine...[47]" Reference 47 doesn't mention tyrosine at all.

Reply: Thanks a lot for the reviewer's valuable comments. According to the reviewer's suggestion, the relevant contents have been rephrased accordingly and the reference has been double-checked and revised according to the contents. (see highlight at the Line 407-412)

Line 407-412 "The colonization and pathogenicity of *Porphyromonas gingivalis*, one of the periodontal pathogens, were regulated by signal transduction pathways based on protein tyrosine phosphorylation and dephosphorylation. In the presence of tyrosine, a tyrosine phosphatase designated Php1 can promote the development and pathogenicity of *Porphyromonas gingivalis*. And in the absence of tyrosine phosphatase

Php1 activity *Porphyromonas gingivalis* was unable to cause periodontal disease in a mouse model of periodontitis[50].”

[50]. Jung YJ, Miller DP, Perpich JD, Fitzsimonds ZR, Shen D, Ohshima J, et al. *Porphyromonas gingivalis* Tyrosine Phosphatase Php1 Promotes Community Development and Pathogenicity. *Mbio*. 2019; 10.

11.Lines 402-404: You say "N,n-dimethylarginine is an endogenous inhibitor of endothelial nitric oxide (NO) synthase, and NO content is reduced in periodontal disease " but reference 51 doesn't mention periodontal disease.

Reply: We appreciate the reviewer’s valuable comments. All the references have been double-checked and revised according to the contents. (see highlight at the Line 431)

Line 431: “N,n-dimethylarginine is an endogenous inhibitor of endothelial nitric oxide (NO) synthase[54] , and NO content is reduced in periodontal disease[55]”

[55]. Andrukhov O, Haririan H, Bertl K, Rausch WD, Bantleon HP, Moritz A, et al. Nitric oxide production, systemic inflammation and lipid metabolism in periodontitis patients: possible gender aspect. *J Clin Periodontol*. 2013; 40: 916-23.

12.Lines 404-406: You say: "Campylobacter concosus is a pathogen of periodontal disease and increases in abundance in periodontal disease, which may also be due to its inhibition of NO production during periodontal inflammation [52]" I assume you're talking about C. concisus but reference 52 is mainly about C. jejuni, C. concisus is only mentioned in passing and that reference doesn't deal with periodontitis or the oral cavity at all. Also C. concisus is not associated with periodontitis per Griffen et al. and Abusleme et al.

Reply: Thanks a lot for the reviewer’s valuable comments. Thank you for providing us with two references. We have read them carefully. The relevant contents have been rephrased as follows:

(1) The “*Campylobacter concosus*” has been corrected to “*Campylobacter concisus*”. (see highlight at the Line 431).

(2) And the reference has been double-checked and revised according to the contents. (see highlight at the Line 432).

(3) According to the reviewer’s suggestion, the relevant contents have been rephrased accordingly. (see highlight at the Line 431-432)

Line 431-432 : “*Campylobacter concisus* is an emerging bacterial pathogen that may play a role in the development of oral inflammatory conditions such as periodontal disease[56,57]”

[56]. Chen J, Liu F, Lee SA, Chen S, Zhou X, Ye P, et al. Detection of IL-18 and IL-1 β protein and mRNA in human oral epithelial cells induced by *Campylobacter concisus* strains. *Biochem Biophys Res Commun*. 2019; 518: 44-9.

[57]. Miljković-Selimović B, Babić T, Kocić B, Aleksić E, Malešević A, Tambur Z. *Campylobacter concisus*. *J*

Thank you very much for the two references you provided, maybe I could not find the relevant description of *Campylobacter concisus* in these two articles. After carefully reviewing the relevant references of *Campylobacter concisus*, the reasons why we regard it as related to periodontitis are as follows:

Campylobacter concisus, an oral commensal, has been reported to be associated with gingivitis and periodontitis. Chen et al. reported that *Campylobacter concisus* upregulated IL-18 and IL-1b in oral epithelial cells, which supported the role of *Campylobacter concisus* in oral inflammatory diseases such as periodontal disease^[10]. Then, a recent study of 16S rRNA gene sequencing of salivary microbes in diabetic patients with periodontitis showed that *Campylobacter concisus* was significantly enriched in chronic periodontitis group^[11]. In addition, Marchesan et al. has reported that microorganisms predominant in periodontal plaque samples such as *Campylobacter concisus* in periodontal plaque samples were significantly associated with human periodontitis^[12]. In conclusion, *Campylobacter concisus* was regarded as the pathogen of periodontal disease in patients with type 2 diabetes mellitus in our manuscript.

13. Figure 1 legend what are the y axis on C and D?

Reply: We appreciate the reviewer’s valuable comments. Figure 1C and 1D legend: Y-axis represents Bray-Curtis dissimilarity. According to the reviewer’s suggestion, the relevant contents have been rephrased accordingly. (see highlight in Fig1C and 1D)

Fig1

14. Figure 3 legend what are the x and y axis of the graphs?

Reply: Thanks a lot for the reviewer’s valuable comments. According to the reviewer’s suggestion, the relevant contents have been rephrased accordingly. (see highlight at line 637-641). Because one figure was added to the front of the manuscript, the original figure 3 was renamed to figure 4.

Line 637-641: “A~D. The X-axis represents \log_2 (Fold Change); Y-axis represents the negative logarithmic conversion of the *P*-value of the difference test between the two groups ($-\log_{10}$ *P*-value). E~H. X-axis enrichment factor (Rich Factor) is the number of differential metabolites annotated to the pathway divided by all identified metabolites annotated to the pathway. Y-axis represents pathways.”

15. I didn't get any legends for the supplemental figures?

Reply: We appreciate the reviewer's valuable comments. According to the reviewer's suggestion, the relevant contents have been rephrased accordingly. (see highlight at line 651-668)

Line 651-668:

“Supplementary Figure Legends

Supplementary Figure 1 PCoA analysis of salivary microbial community between T2DM group and normal control group. Data points in different shapes and colors represent for samples in variant conditions. Scales of X-axis and Y-axis are the projection of samples' coordinates in 2-dimension. PCoA axis1 and PCoA axis2 stand for the possible factors that drive the changes of microbiota structure in groups, which needs to be interpreted in combination with groups features.

Supplementary Figure 2 PCoA analysis of supragingival plaque microbial community between T2DM group and normal control group. Data points in different shapes and colors represent for samples in variant conditions. Scales of X-axis and Y-axis are the projection of samples' coordinates in 2-dimension. PCoA axis1 and PCoA axis2 stand for the possible factors that drive the changes of microbiota structure in groups, which needs to be interpreted in combination with groups features.

Supplementary Figure 3 KEGG metabolic pathway enrichment map of salivary microorganisms between T2DM group and normal control group. X-axis shows reporterscore values, small blocks on Y-axis show pathways. Differences are significant when bars exceed dashed lines.

Supplementary Figure 4 KEGG metabolic pathway enrichment map of supragingival plaque microorganisms between T2DM group and normal control group. X-axis shows reporterscore values, small blocks on Y-axis show pathways. Differences are significant when bars exceed dashed lines.”

16. Please provide a link to the analysis code used.

Reply: Thanks a lot for the reviewer's valuable. The analysis code used are *R* package *pcaMethods*, *R* package *pls*, *R* package *ggplot2*.

Minor points

17. Line 138: I believe the program is MetaGeneMark, not MetaGeneMarker.

Reply: Thanks a lot for the reviewer's valuable comments. According to the reviewer's suggestion, the relevant contents have been rephrased accordingly. The “*MetaGeneMarker*” has been corrected to

“*MetaGeneMark*”. (see highlight at line 145)

18.Line 189: the program is Bowtie2, not Botwie2.

Reply: We appreciate the reviewer’s valuable comments. According to the reviewer’s suggestion, the relevant contents have been rephrased accordingly. The “*Botwie2*” has been corrected to “*Bowtie2*”. (see highlight at line 196)

19.Lines 230, 231: where you say "species level branches" and "bacteria", what the figure is depicting is actually taxonomic ranks that can be different levels species or above.

Reply: Thanks a lot for the reviewer’s valuable comments. According to the reviewer’s suggestion, the relevant contents have been rephrased accordingly. (see highlight at line 236, 237)

Line 236, 237: “LEfSe analysis demonstrated that these biomarkers were significantly different between the two groups.”

The “*bacteria*” has been corrected to “*biomarkers*”. (see highlight at line 238 and 243)

20.Line 241: not "fecal" bacteria.

Reply: We appreciate the reviewer’s valuable comments. According to the reviewer’s suggestion, the “*fecal*” has been corrected to “*oral*”. (see highlight at line 249)

21.Lines 245 and 246 and figures - those abbreviations TS, CS, etc are never defined and I think it would be easier to understand with T2DM saliva or control saliva written out.

Reply: Thanks a lot for the reviewer’s valuable comments. According to the reviewer’s suggestion, the “*TS*” has been corrected to “*T2DM saliva*” in line 258. (see highlight at line 258).The “*CS*” has been corrected to “*control saliva*” in line 259. (see highlight at line 259). The “*TP*” has been corrected to “*T2DM supragingival plaque*” in line 260. (see highlight at line 260). The “*CP*” has been corrected to “*control supragingival plaque*” in line 261. (see highlight at line 261). In addition, when drawing these figures, because the size of the figure is limited, those abbreviations TS, CS, etc are defined. In addition, in order to make those abbreviations more intuitive, we have added an explanation of those abbreviations to each figure legend.

Line 257- 261: “Based on the threshold values Reporterscore > 1.65, 32 KEGG pathways were significantly enriched in T2DM saliva, and 4 KEGG pathways were significantly enriched in control saliva (Supplementary Fig. 3). Twenty-eight KEGG pathways were significantly enriched in T2DM supragingival plaque, and 36 KEGG pathways were significantly enriched in control supragingival plaque (Supplementary Fig. 4).”

Line 619-621; 625-627; 631-633; “TS refers to the saliva samples of T2DM group. TP refers to the supragingival plaque samples of T2DM group. CS refers to the saliva samples of control group. CP refers to

the supragingival plaque samples of control group.”

Griffen, Ann, L, Beall, Clifford, J, Campbell, James, H, Firestone, Noah, D, Kumar, Purnima, S, Yang, Zamin, K, Podar M, Leys, Eugene, J. Distinct and complex bacterial profiles in human periodontitis and health revealed by 16S pyrosequencing. *ISME J [Internet]*. 2012 Jun;6(6):1176-85. Available from: <http://www.ncbi.nlm.nih.gov/pubmed/22170420>

Abusleme L, Dupuy AK, Dutzan N, Silva N, Bureson JA, Strausbaugh LD, Gamonal J, Diaz PI. The subgingival microbiome in health and periodontitis and its relationship with community biomass and inflammation. *ISME J [Internet]*. 2013;7(5):1016-25.

Reviewer #2 (Comments for the Author):

General comment

1. The authors examined saliva and supragingival plaque samples using metagenomic and metabolic analyses and found the differences in oral bacterial and metabolite compositions of patients with T2DM from those of healthy control. Particularly, it is interesting that the focus is on subjects who have not suffered any oral diseases. The present study could provide new perspectives for understanding the involvement of diabetes in increasing risk of periodontal disease.

Reply: We appreciate the reviewer's support and valuable comments.

2. My major concern is the lack of details on oral conditions of subjects. The authors emphasize that the compositions of oral microbiota and metabolites change in T2DM patients even if the oral condition is healthy, and thus T2DM increases the risk of periodontal disease. However, the criteria of "healthy oral condition" in the present study is quite ambiguous. The authors only describe that subjects with caries or periodontal diseases are excluded, on line 114-116. The oral conditions are known to greatly influence on bacterial composition of oral microbiota. For instance, deepening of the periodontal pocket increases anaerobic environment in oral cavity and enriches anaerobic bacteria, even if the periodontal condition is within a healthy range. Therefore, it is possible that the present results just reflect differences in periodontal conditions rather than T2DM. In fact, the present results (increases of *P. gingivalis* or cadaverine in T2DM) are closely similar to alteration in oral cavity of subjects with deep periodontal pockets. Thus, it seems difficult to conclude T2DM increases periodontal disease risk via alteration of oral microbiota from the present results.

Reply: Thanks a lot for the reviewer's valuable comments and suggestions. The comments and suggestions are very helpful for revising and improving our manuscript, as well as the important guiding significance to our study. The subjects we chose were actually oral health. An experienced dentist examined the oral cavity of the subjects in our study. The number of decayed, missing, or filled teeth (DMFT) was recorded with the exception of wisdom teeth. Gingival and periodontal tissues were examined in a logical order according to the Community Periodontal Index[19]. First, the gingival tissues were examined with a visual inspection to assess (somewhat subjectively) the color and swelling of the tissues. Then, gingival index (GI), bleeding index (BI), clinical attachment loss (CAL), plaque index (PLI) and probing depth (PD) were measured by a KPC15 probe (Kangqiao, Shanghai, China). The results of periodontal examination showed that there was no significant difference in periodontal pocket depth between the two groups. The relevant contents in the *Materials and methods* section have been added accordingly in the revised manuscript. (see highlight at the

end of the *Materials and methods* section in line 117-123). The relevant results have been added accordingly in Table 1.

Line 117-123: “An experienced dentist examined the oral cavity of the subjects in our study. The number of decayed, missing, or filled teeth (DMFT) was recorded with the exception of wisdom teeth. Gingival and periodontal tissues were examined in a logical order according to the Community Periodontal Index[19]. First, the gingival tissues were examined with a visual inspection to assess (somewhat subjectively) the color and swelling of the tissues. Then, gingival index (GI), bleeding index (BI), clinical attachment loss (CAL), plaque index (PLI) and probing depth (PD) were measured by a KPC15 probe (Kangqiao, Shanghai, China).”

[19].Eke PI, Page RC, Wei L, Thornton-Evans G, Genco RJ. Update of the case definitions for population-based surveillance of periodontitis. *J Periodontol.* 2012; 83: 1449-54.

Table 1:

Table 1 Demographic and clinical characteristics of all participants

Characteristics	T2DM patients	Healthy controls	p value
Number	10	10	-
Male/female	7/3	5/5	0.361
Age (years)	44.7 ± 11.67	42.20 ± 11.27	0.632
Height (cm)	1.71 ± 0.07	1.68 ± 0.08	0.422
Weight (kg)	76.3 ± 11.72	61.80 ± 7.32	*0.005
BMI (kg/m ²)	25.97 ± 2.33	21.84 ± 1.82	*0.000
DMFT	0	0	-
GI	0	0	-
PLI	0.80±0.63	0.70±0.48	0.696
PD(mm)	2.48 ±0.12	2.47±0.13	0.859
BI	0	0	-
CAL(mm)	0	0	-

Values are expressed as the mean ± standard deviation. Significant differences are indicated by **p* < 0.05. BMI, body mass index.

Specific comment

3.Result: To make it easier for the reader to understand the overall bacterial composition at genus or species level, the author should additionally draw figures showing bacterial composition in each subject (e.g. barplot).

Reply: We appreciate the reviewer’s valuable comments. According to the reviewer’s suggestion, the relevant contents and figures in the *Materials and methods* section and *Results* section have been added accordingly in the revised manuscript. (see highlight at the end of the *Materials and methods* section in line 252-255, Figure Legends in line 622-627)

Line 252-255 “In order to describe the taxonomic profiles of the microbial communities more intuitively, taxonomy barplots at genus level (Fig. 2A, Fig. 2C) and species level (Fig. 2B, Fig. 2D) were plotted for each sample. Top 30 taxonomies were plotted from sorted abundance table when more than 30 taxonomies are annotated.”

Line 622-627: **Fig. 2 Taxonomic profiles of the microbial communities at the genus level and species level in each sample.** (A) salivary bacteria at the genus level; (B) salivary bacteria at the species level; (C) supragingival plaque bacteria at the genus level; (D) supragingival plaque bacteria at the species level. TS refers to the saliva samples of T2DM group. TP refers to the supragingival plaque samples of T2DM group. CS refers to the saliva samples of control group. CP refers to the supragingival plaque samples of control group.

Fig. 2

4.Lines 276-307: It is hard to interpret these results from figure 3-4 and supplementary tables. For instance, which is the result of cadaverine? The authors should redraw figures to clarify it.

Reply: Thanks a lot for the reviewer’s valuable comments. Because there are too many relevant results

involved, after careful consideration, we added the specific results to the supplementary table. According to the reviewer's suggestion, the relevant contents have been rephrased accordingly.

In order to demonstrate more clearly the metabolites with significant differences ($VIP > 1.5$, $P < 0.05$) in our study, including cadaverine, they have been placed separately in the table S1 and S2.

In order to more intuitively observe the results of the correlation analysis, the names of the relevant metabolites and the related P values have been added separately to the table S5.

5.Lines 298-300: This sentence is unclear. Please rephrase.

Reply: We appreciate the reviewer's valuable comments. According to the reviewer's suggestion, the relevant contents have been rephrased accordingly. (see highlight in Line 311-313)

Line 311-313: "Specifically, some periodontal pathogens such as *Parvimonas micra*, *Porphyromonas gingivalis*, and *Treponema denticola* were significantly associated with 15, 32, and 40 salivary metabolites, respectively ($P < 0.05$)."

6.Lines 354-358: These sentences are complicated. If a carbohydrate limited-diet is less likely to cause dental caries than a normal diet as described, doesn't the risk of dental caries decrease in T2DM patients?

Reply: Thanks a lot for the reviewer's valuable comments. According to the reviewer's suggestion, the relevant contents have been rephrased accordingly. (see highlight in Line 369-377)

Line 369-377: "Therefore, our study preliminarily indicated that there was no significant difference in the risk of dental caries between T2DM patients and normal controls. We suspect that the reasons may be various. On the one hand, that restriction of ingestion of refined carbohydrates reduced caries in patients with T2DM. On the other hand, a high salivary glucose level[45] or hyposalivation[46] in T2DM might increase the caries risk. Therefore, it may be these contradictory associations that lead to no significant change in oral cariogenic bacteria in patients with T2DM. As for whether the actual incidence of dental caries in patients with type 2 diabetes will change, we believe that longitudinal trials with a larger sample size are needed to investigate."

[45]. Mascarenhas P, Fatela B, Barahona I. Effect of diabetes mellitus type 2 on salivary glucose--a systematic review and meta-analysis of observational studies. *Plos One*. 2014; 9: e101706.

[46]. Marques R, Da SJ, Vieira LC, Stefani CM, Damé-Teixeira N. Salivary parameters of adults with diabetes mellitus: a systematic review and meta-analysis. *Oral Surg Oral Med Oral Pathol Oral Radiol*. 2022; 134: 176-89.

7.Table S1, S2, and S5: Please show compound names.

Reply: We appreciate the reviewer's valuable comments. According to the reviewer's suggestion, the compound names in table S1, S2, and S5 have been added accordingly. (see highlight in table S1, S2, and S5)

References:

- [1] Han Y W. *Fusobacterium nucleatum*: a commensal-turned pathogen[J]. *Curr Opin Microbiol*, 2015,23:141-147.
- [2] Bolstad A I, Jensen H B, Bakken V. Taxonomy, biology, and periodontal aspects of *Fusobacterium nucleatum*[J]. *Clin Microbiol Rev*, 1996,9(1):55-71.
- [3] Moore W E, Moore L V. The bacteria of periodontal diseases[J]. *Periodontol 2000*, 1994,5:66-77.
- [4] Chaushu S, Wilensky A, Gur C, et al. Direct recognition of *Fusobacterium nucleatum* by the NK cell natural cytotoxicity receptor NKp46 aggravates periodontal disease[J]. *PLoS Pathog*, 2012,8(3):e1002601.
- [5] Polak D, Wilensky A, Shapira L, et al. Mouse model of experimental periodontitis induced by *Porphyromonas gingivalis*/*Fusobacterium nucleatum* infection: bone loss and host response[J]. *J Clin Periodontol*, 2009,36(5):406-410.
- [6] Kang W, Jia Z, Tang D, et al. *Fusobacterium nucleatum* Facilitates Apoptosis, ROS Generation, and Inflammatory Cytokine Production by Activating AKT/MAPK and NF- κ B Signaling Pathways in Human Gingival Fibroblasts[J]. *Oxid Med Cell Longev*, 2019,2019:1681972.
- [7] Ford R C, Beis K. Learning the ABCs one at a time: structure and mechanism of ABC transporters[J]. *Biochem Soc Trans*, 2019,47(1):23-36.
- [8] Theodoulou F L, Kerr I D. ABC transporter research: going strong 40 years on[J]. *Biochem Soc Trans*, 2015,43(5):1033-1040.
- [9] Gao L, Ma Y, Li X, et al. Research on the roles of genes coding ATP-binding cassette transporters in *Porphyromonas gingivalis* pathogenicity[J]. *J Cell Biochem*, 2020,121(1):93-102.
- [10] Chen J, Liu F, Lee S A, et al. Detection of IL-18 and IL-1 β protein and mRNA in human oral epithelial cells induced by *Campylobacter concisus* strains[J]. *Biochem Biophys Res Commun*, 2019,518(1):44-49.
- [11] Lu C, Zhao Q, Deng J, et al. Salivary Microbiome Profile of Diabetes and Periodontitis in a Chinese Population[J]. *Front Cell Infect Microbiol*, 2022,12:933833.
- [12] Marchesan J, Jiao Y, Schaff R A, et al. TLR4, NOD1 and NOD2 mediate immune recognition of putative newly identified periodontal pathogens[J]. *Mol Oral Microbiol*, 2016,31(3):243-258.

December 12, 2022

Prof. Yu Tian
Department of Operative Dentistry and Endodontics, School of Stomatology, the Fourth Military Medical University
Xi'an
China

Re: Spectrum03796-22R1 (Dysbiosis of oral microbiota and metabolite profiles associated with type 2 diabetes mellitus)

Dear Prof. Yu Tian:

Your manuscript has been accepted, and I am forwarding it to the ASM Journals Department for publication. You will be notified when your proofs are ready to be viewed.

Sincerely,

Justin Kaspar
Editor, Microbiology Spectrum

Journals Department
Supplemental Table 4: Accept
Supplementary Table2: Accept
Supplementary Table3: Accept
Supplemental Material: Accept
Supplemental Table 5: Accept
Supplementary Table1: Accept